# Outcomes of Radiotherapy for Mesenchymal and Non-Mesenchymal Subtypes of Gastric Cancer

**DOI:** 10.3390/cancers12040943

**Published:** 2020-04-10

**Authors:** Jeong Il Yu, Hee Chul Park, Jeeyun Lee, Changhoon Choi, Won Ki Kang, Se Hoon Park, Seung Tae Kim, Tae Sung Sohn, Jun Ho Lee, Ji Yeong An, Min Gew Choi, Jae Moon Bae, Kyoung-Mee Kim, Heewon Han, Kyunga Kim, Sung Kim, Do Hoon Lim

**Affiliations:** 1Departments of Radiation Oncology, Samsung Medical Center, Sungkyunkwan University School of Medicine, 81 Irwon-ro, Gangnam-gu, Seoul 06351, Korea; ro.yuji651@gmail.com (J.I.Y.); changhoon1.choi@samsung.com (C.C.); dh8.lim@samsung.com (D.H.L.); 2Department of Medical Device Management and Research, Samsung Advanced Institute for Health Sciences and Technology, Sungkyunkwan University, Seoul 06351, Korea; 3Division of Hematology-Oncology, Department of Medicine, Samsung Medical Center, Sungkyunkwan University School of Medicine, 81 Irwon-ro, Gangnam-gu, Seoul 06351, Korea; wonki.kang@samsung.com (W.K.K.); sh1767.park@samsung.com (S.H.P.); seungtae1.kim@samsung.com (S.T.K.); 4Departments of Surgery, Samsung Medical Center, Sungkyunkwan University School of Medicine, Seoul 06351, Korea; ts.sohn@samsung.com (T.S.S.); junho3371.lee@samsung.com (J.H.L.); jar319.an@samsung.com (J.Y.A.); mingew.choi@samsung.com (M.G.C.); jmoon.bae@samsung.com (J.M.B.); 5Departments of Pathology, Samsung Medical Center, Sungkyunkwan University School of Medicine, Seoul 06351, Korea; km7353.kim@samsung.com; 6Statistics and Data Center, Research Institute for Future Medicine, Samsung Medical Center, Seoul 03181, Korea; heewon818@gmail.com (H.H.); kyunga.j.kim@samsung.com (K.K.); 7Department of Surgery, Samsung Changwon Hospital, Sungkyunkwan University School of Medicine, Changwon 06351, Korea; sungkimm@skku.edu

**Keywords:** adjuvant therapy, gastrointestinal tract, genetic diagnosis, radiosensitivity

## Abstract

*Background:* The purpose of this study was to evaluate the clinical outcomes following postoperative chemotherapy (XP) versus chemoradiotherapy (XP-RT) according to mesenchymal subtype based on RNA sequencing in gastric cancer (GC) in a cohort of the Adjuvant chemoRadioTherapy In Stomach Tumor (ARTIST) trial. Methods: Of the 458 patients enrolled in the ARTIST trial, formalin-fixed, paraffin-embedded (FFPE) specimens were available from 106 (23.1%) patients for RNA analysis. The mesenchymal subtype was classified according to a previously reported 71-gene MSS/EMT signature using the NanoString assay. Results: Of the 106 patients analyzed (50 in XP arm, 56 in XP-RT arm), 36 (34.0%) patients were categorized as mesenchymal subtype by NanoString assay. Recurrence-free survival (RFS, p = 0.009, hazard ratio (HR) = 2.11, 95% confidence interval (CI): 1.21–3.70) and overall survival (OS, p = 0.003, HR = 2.28, 95% CI: 1.31–3.96) were significantly lower in the mesenchymal subtype than in the non-mesenchymal subtype. In terms of post-operative radiotherapy (RT), mesenchymal subtype was not an independent variable to predict RFS or OS regardless to the assigned arm (XP with or without RT) in this patient cohort. However, there was a trend in the adjuvant XP arm, which showed higher OS than the XP-RT arm for the mesenchymal subtype and lower OS than the XP-RT arm for the non-mesenchymal subtype. *Conclusions:* We could not determine any significant differences between the mesenchymal and non-mesenchymal subtypes with respect to the effects of adjuvant XP with or without RT in gastric cancer following curative surgery.

## 1. Introduction

Gastric cancer (GC) remains an unresolved major health problem in the world, ranking fifth among the most common cancers, with an estimated incidence of 951,000 cases in 2012 [1]. Furthermore, it is the third leading cause of cancer-related deaths, with 723,100 patients dying because of it in 2012. Nevertheless, a gradual improvement in GC management has been achieved, based on pathophysiological studies, novel surgical techniques, and/or emerging systemic therapies.

Despite significant improvements in clinical outcomes related to the implementation of screenings, advances in surgical techniques and/or usage of (neo-)adjuvant chemotherapy and/or concurrent chemoradiotherapy, a considerable proportion of patients still experience recurrences and die of GC [2,3,4].

With the recent development of genetic analysis techniques, the molecular classification of GC and the prognostic value of the different subtypes have been actively studied. The Cancer Genome Atlas Research Network suggested four types of GC molecular classes [5], whereas the Asian Cancer Research Group (ACRG) proposes another classification [6]. In particular, the ACRG suggested prognostic differences according to each molecular subtype and validated the survival differences in an independent cohort. According to this study, the mesenchymal (microsatellite-stable with epithelial-to-mesenchymal transition phenotype, MSS/EMT) tumors showed the worst prognosis with younger age at diagnosis, and higher recurrence rate with first presentation of peritoneal seeding. In addition to these clinical features of the mesenchymal subtype, the association of hypoxia and the increase in poly [adenosine diphosphate-ribose] polymerase-1 (PARP-1), which repairs deoxyribonucleic acid (DNA) damage, has been reported [7,8,9].

Recently, our group performed a study to predict outcomes in patients with the mesenchymal subtype using a NanoString assay in 70 ACRG specimens [10]. The mesenchymal subtype showed significantly worse survival compared to the non-mesenchymal subtype following curative surgery in GC [11,12,13]. The impact of mesenchymal subtype which could be clearly related with radioresistance, like hypoxia and increment of PARP-1, in terms of radiotherapy (RT) efficacy has not been defined yet although several studies have demonstrated that mesenchymal subtype predicts poor outcome following chemotherapy in GC [11,12,13]. It would be appropriate to evaluate the efficacy of RT on mesenchymal subtype of GC through the ARTIST trial which had been conducted to compare postoperative chemotherapy (XP) versus chemoradiotherapy (XP-RT) following complete curative resection with D2 lymph node dissection in GC (clinical trials.gov identifier NCT00323830).

In this study, we investigated the clinical outcomes according to the application of adjuvant XP-RT or XP for mesenchymal and non-mesenchymal subtype cancers in a cohort of the ARTIST trial. 

## 2. Methods

### 2.1. Patients and Samples

This study was performed on patients who participated in the ARTIST trial who agreed to have their tissue studied, and whose tissue surgical specimens were available and sufficient for ribonucleic acid (RNA) extraction. A total of 458 patients (228 assigned to XP and 230 to XP-RT arm), who had received curative D2 resection without preoperative treatment, were enrolled in the randomized phase III study, ARTIST trial. The XP arm was expected to receive six cycles of XP regimen (capecitabine 1000 mg/m^2^ twice daily on days 1 to 14; cisplatin 60 mg/m^2^ on day 1 every 3 weeks) and the XP-RT arm would receive 25 fractions of 45 gray (Gy) RT with capecitabine (825 mg/m^2^ twice daily) after two cycles of XP (as in the XP arm) followed by two additional cycles of planned XP. The planned treatment was completed in 75.4% of patients in the XP arm and 81.7% in the XP-RT arm [11]. Other details of the ARTIST trial, including chemotherapy and RT protocols, were described in previous reports [8,9,10]. From the 458 patients enrolled, 106 tissue samples of 359 patients except 99 patients with stage I were available for evaluation in the present study. Of these, 56 patients were in the XP-RT arm and the remaining 50 patients were in the XP arm (Figure 1).

### 2.2. Mesenchymal Gene Signature

Details about the development of the 71-gene MSS/EMT signature (consisting of 60 upregulated and 11 downregulated genes) using the NanoString assay, and the validation procedure using the conventional Affymetrix method, are described in a previous report [10]. In that study, 73 samples from the ARTIST cohort were tested using the 71-gene MSS/EMT signature to validate their mesenchymal subtype. Twenty out of 73 samples were classified as mesenchymal subtype tumors, which is equivalent to the MSS/EMT subtype in terms of their dismal outcome, typical characteristics of whole stomach involvement, poorly-differentiated or signet ring cell carcinoma, and low microsatellite instability.

In this study, we used the outcomes of previous study evaluating mesenchymal and non-mesenchymal subtype using MSS/EMT gene signature analysis using the NanoString assay in a total of 106 samples, which consisted of 73 samples used as a validation set and 33 additional specimens from patients of the ARTIST trial evaluated through further work after publication of the previous study. 

The first site of recurrence was used for the classification and/or analysis of recurrence. Simultaneous recurrence was defined as any recurrence detected within 2 weeks after the first detection of recurrence. Loco-regional recurrence (LRR) was defined as recurrence at one of following sites: anastomosis area, remnant stomach, tumor bed, duodenal stump, or regional lymph nodes (LN) within the RT field in the XP-RT group or the hypothetical RT field in the XP group. All cases of suspected LRR were reviewed and evaluated by dedicated radiation oncologists (JIY and DHL; specialists in gastrointestinal tumors, including stomach cancer) as described in a previous study [13].

### 2.3. Ethical Approval and Informed Consent Statement

The authors stated that all methods of this study were carried out in accordance with the Declaration of Helsinki, and the protocol for the present study was reviewed and approved by the Samsung Medical Center Institutional Review Board (IRB No. 2010-12-088) and all participants in the ARTIST trial consented to this study after being informed about the purpose and investigational nature. 

### 2.4. Statistical Analysis

Baseline characteristics were compared between the mesenchymal and non-mesenchymal subtypes, using chi-square or Fisher’s exact test for categorical variables and Student’s t-test or Mann–Whitney U-test for continuous variables, as appropriate. For each survival-related event, such as LRR, recurrence, or death, survival times were calculated from the date of surgery to the date of event detection, or the date of the last follow-up visit. LRR-free survival (LRRFS), recurrence-free survival (RFS) and overall survival (OS) curves were estimated and compared between the XP and XP-RT groups, both for mesenchymal and non-mesenchymal subtypes, with the adjustment for stage, operation type and classification of Lauren based on the multivariate Cox proportional hazards model. Statistical analysis was performed using SAS software version 9.4 (SAS Institute Inc., Cary, NC, USA) and *p <* 0.05 was considered statistically significant.

## 3. Results

### 3.1. Patients

Among the 458 patients who participated in the ARTIST trial, a total of 106 (23.1%) formalin-fixed, paraffin-embedded (FFPE) samples that were available for targeted profiling by the NanoString nCounter assay were evaluated and analyzed in the present study. The baseline characteristics of patients enrolled in the present study and all patients in the ARTIST trial are shown in Appendix A. Patients enrolled in this study had significantly more metastatic lymph nodes, and lymphovascular invasion (LVI) and perineural invasion (PNI) were more common than those of all the patients in the ARTIST trial. Clinical-Trials.gov identifier: NCT0176146. Trial Registration: clinical trials.gov identifier NCT00323830 (date of registration: May 10, 2006).

Among the 106 patients enrolled in this study, 56 were assigned to the XP-RT group and the remaining 50 patients were assigned to the XP group. When testing molecular subtypes using the developed MSS/EMT signature, 36 out of 106 patients were classified as having the mesenchymal subtype.

Table 1 displays the detailed characteristics of the enrolled patients having mesenchymal or non-mesenchymal subtypes of GC. In the mesenchymal subtype group, the ratio of diffuse-type GC, classified according to Lauren, was significantly higher than that of the non-mesenchymal subtype group (88.9% versus 60.0%, *p* = 0.004). Additionally, the proportion of patients who received total gastrectomy was higher in the mesenchymal subtype group (55.6% versus 40.0%, *p* = 0.128), although no statistical significance was found. No differences in the percentage of patients receiving XP or XP-RT as adjuvant treatment, nor significant differences in the pathologic staging, were found between the two subtypes.

Baseline characteristics of patients randomly assigned to the XP-RT and XP arms of the ARTIST trial are displayed in Appendix A. Age was slightly but significantly lower in the XP-RT arm than in the XP arm. 

### 3.2. Patterns of Recurrence

During follow up (median: 43.6 months, range: 1.8–72.0 months), recurrence was identified in 50 patients (47.2%). Among them, 20 (57.1%) recurrences had developed in the mesenchymal subtype group and the remaining 30 (42.3%) recurrences occurred in the non-mesenchymal subtype group. Median time to any recurrence was 12.9 months (range: 3.8–75.1 months) in the mesenchymal subtype group and 18.0 months (range: 4.5–81.3 months) in the non-mesenchymal subtype group. LRR developed in 20 out of 50 patients, and seven of them were classified as mesenchymal subtype. Median time to LRR was 26.9 months (range: 3.8–81.3 months) in the mesenchymal subtype group and 23.7 months (range: 1.77–154.9 months) in the non-mesenchymal subtype group. Detailed patterns of recurrence according to subtype and adjuvant treatment are shown in Appendix A. Peritoneal seeding was the major recurrence pattern in the XP-RT arm, especially in the mesenchymal subtype, and LRR was more frequent in the XP arm. Fifty-one deaths were registered during this period (30 in the non- mesenchymal and 21 in the mesenchymal subtype group).

### 3.3. Prognostic Factors and Survival Outcomes

Table 2 shows the univariate analysis outcomes of LRRFS, RFS and OS according to the variables, including mesenchymal versus non-mesenchymal subtypes. LRRFS was not different for the mesenchymal and non-mesenchymal subtypes (*p* = 0.275); PNI was the only significant prognostic factor for LRRFS (*p* = 0.044, hazard ratio (HR) = 3.57, 95% confidence interval (CI): 1.03–12.20). RFS was significantly lower in the mesenchymal subtype group (*p* = 0.009, HR = 2.11, 95% CI: 1.21–3.70). Other significant prognostic factors for RFS were type IV of the macroscopic type (*p* < 0.001), total gastrectomy (*p* = 0.007), stage IV (*p* < 0.001), PNI (*p* = 0.014), and Lauren classification other than diffuse-type (*p* = 0.028). The RFS was not different according to the adjuvant treatment of XP or XP-RT (*p* = 0.500). OS was also significantly lower in the mesenchymal subtype group (*p* = 0.003, HR = 2.28, 95% CI: 1.31–3.96). Other significant prognostic factors for OS were: type IV of the macroscopic type (*p* < 0.001), PNI (*p* = 0.034), total gastrectomy (*p* = 0.002), and stage IV (*p* < 0.001). Adjuvant treatment of XP or XP-RT did not affect OS. 

### 3.4. Survival Outcomes for Mesenchymal and Non-Mesenchymal Subtypes

To evaluate the different roles of adjuvant XP-RT or XP between mesenchymal and non-mesenchymal subtypes, multivariate analysis in both mesenchymal and non-mesenchymal subtype was performed including the following factors, which were identified as significant factors not only in the present study but also in many other studies: stage, type of operation, and Lauren classification. Table 3 shows the outcomes of multivariate analyses of RFS and OS for mesenchymal and non-mesenchymal subtypes. Additional results of multivariate analyses of RFS and OS for mesenchymal and non-mesenchymal subtypes including another well-known prognostic factor of PNI are presented in Appendix A.

In the multivariate analysis of RFS for the mesenchymal subtype, stage and total gastrectomy were significant prognostic factors, compared to the non-mesenchymal subtype, which was significantly affected by stage, and type of Lauren classification. As displayed in Figure 2A,B, adjusted RFS was not significantly different according to the adjuvant treatment of XP or XP-RT, either for the mesenchymal or non-mesenchymal subtype. Forest plots were used to represent adjusted RFS with HR and 95% CI for the mesenchymal and non-mesenchymal subtypes (Figure 3).

No significant prognostic factors were found in the multivariate analysis of OS for the mesenchymal subtype. On the contrary, stage, total gastrectomy, and type of Lauren classification were significant prognostic factors of OS for the non-mesenchymal subtype. OS was not significantly different according to the adjuvant treatment of XP or XP-RT, either for the mesenchymal or non-mesenchymal subtype. As displayed in Figure 2C,D, however, there was a minor difference in OS between mesenchymal and non-mesenchymal subtypes according to the adjuvant treatment group. The adjuvant XP group showed slightly higher adjusted OS than the XP-RT group for the mesenchymal subtype, and lower adjusted OS than the XP-RT group for the non-mesenchymal subtype. Forest plots were used to represent adjusted OS with HR and 95% CI for the mesenchymal and non-mesenchymal subtypes (Figure 3).

## 4. Discussion

In the present study we evaluated the clinical outcomes according to the adjuvant treatment (XP-RT or XP) for mesenchymal and non-mesenchymal subtypes of GC in a cohort of the ARTIST trial. The mesenchymal subtype showed worse prognostic factors, such as frequent recurrence and lower OS, than the non-mesenchymal subtype, as shown in other reports. However, we could not find any differences between the mesenchymal and non-mesenchymal subtypes in terms of OS as well as RFS.

Beyond the previous histologic and/or anatomic classification of cancer [14,15,16], there is a growing body of research and evidence supporting the use of molecular analysis for precise classification and tailored management [5,6,17,18]. Although many advances have been made in the management of GC, it still has a poor prognosis in advanced tumors, becoming one of the areas where molecular classification is being actively attempted. Our group, which is the third referral institution of Korea, proposed four subtypes of molecular classification linked to recurrence patterns and prognosis, as well as to distinct patterns of genomic alterations based on ACRG data which have been validated in institutional cohorts [6]. 

Among the four subtypes of our molecular classification, the mesenchymal-like type, which showed loss of expression of the *CDH1* which encodes the protein E-cadherin. It is known that loss of E-cadherin function decreases the power of cell-to-cell adhesion and increases the cellular motility [19]. It is suggested that the efficacy of local treatment including RT could be limited, because of high metastatic potential originated from these characteristics of the mesenchymal-like type. Actually, it is associated with a younger age of occurrence and is diagnosed as diffuse-type at stage III/IV, showed a significantly higher recurrence rate, higher probability of developing peritoneal seeding at the first site of recurrence, and extremely poor survival compared to other subtypes. Furthermore, it has been reported that PARP-1, which is involved in the mechanism of radiation induced DNA damage repair, is increased in the mesenchymal phenotype, and it could reduce the cell killing effects of RT in prostate cancer [7,8]. In addition, hypoxia, which is a hallmark of tumor and the most important cause of radioresistance, was reported to be associated with the loss of E-cadherin [9].

Our group also proposed the classification of certain GC as mesenchymal type by means of a targeted NanoString gene expression profile [10]. This unique subtype classification may positively impact the standard management of GC, promoting modifications on current treatment protocols which would improve their clinical outcomes.

RT is one of the main therapeutic modalities in the oncology fields. Though adjuvant RT combined with chemotherapy showed survival advantages over surgery alone in the Intergroup trial 0116 [20,21], there is still controversy as to the real efficacy of RT as an adjuvant modality in complete D2-resection GC [22]. The ARTIST trial was a randomized phase III trial designed to evaluate the advantages of adjuvant XP-RT over adjuvant XP after complete D2-resection in GC [11,12,13]. Superiority of adjuvant XP-RT over XP was not detected in these patients, except for those with LN metastasis.

Although the optimal indications of adjuvant XP-RT remain controversial, RT is still one of the most valuable and important treatment modalities in the management of GC, especially in the neoadjuvant or palliative setting. Furthermore, there is a rapid development of RT technology [23], also accompanied by an increased understanding of radiation biology [24,25]. In this respect, evaluating and comparing the effects of RT on GC of the mesenchymal subtype is of paramount importance, considering its heterogeneity and difficult therapeutic management.

The mesenchymal subtype is known to be closely related to younger age of occurrence and diffuse type of the Lauren classification, showing a worse prognosis and higher recurrence rate compared with other subtypes [6,17,18]. Those characteristics of the mesenchymal subtype were observed in the present study as well. Our group has continued to conduct studies to screen out which GC patients might benefit from adjuvant XP-RT over XP [4,11,12,13,22]. We found and reported that the benefit of adjuvant XP-RT over XP in GC is reduced to patients with diffuse-type GC and younger age, which correspond to characteristics of the mesenchymal type [12].

The effect of adjuvant XP-RT in mesenchymal subtype of GC in the present study was not significantly different than that of non-mesenchymal subtype in terms of RFS, in contrast to the possibility that the effect of RT might be reduced when considering the characteristics of this subtype. On the other hand, the adjuvant XP-RT group showed a lower OS curve than the XP group for the mesenchymal subtype, although without statistical significance. Although we failed to detect significant difference of RFS between the subtypes according to the adjuvant XP-RT and XP, this marginal difference of OS might be related to the different patterns of recurrence between the subtypes, especially LRR and/or peritoneal seeding. Therefore, the possibility of poorer outcomes after adjuvant XP-RT than after XP for the mesenchymal subtype with complete D2-resection cannot be ruled out. Further research on this issue is needed.

The present study has some important limitations. First, this study was evaluated in a fraction of the participants in the ARTIST trial, mainly because of availability of tissue specimens. Therefore, it is not possible to avoid selection bias in the XP-RT and XP groups, which is minimized by the random allocation nature of phase III. Second, the patients enrolled in this study have characteristics that differ from those of all patients in the ARTIST trial. Third, there could be problems originating from variation in outcomes due to ethnic differences, since the ARTIST trial was conducted at a single Korean institution. Further similar studies are essential for validation in other ethnicities in order to generalize present outcomes, because it is well-known that the characteristics and/or clinical outcomes of gastric cancer are quite distinct according to ethnicity [26,27,28].

## 5. Conclusions

We could not determine any significant differences on the effect of adjuvant XP-RT on RFS between mesenchymal and non-mesenchymal subtypes in this cohort of the ARTIST trial. There was a minor difference, however, in the adjuvant XP-RT group showing lower OS than the XP group for the mesenchymal subtype and higher OS for the non-mesenchymal subtype.

## Figures and Tables

**Figure 1 cancers-12-00943-f001:**
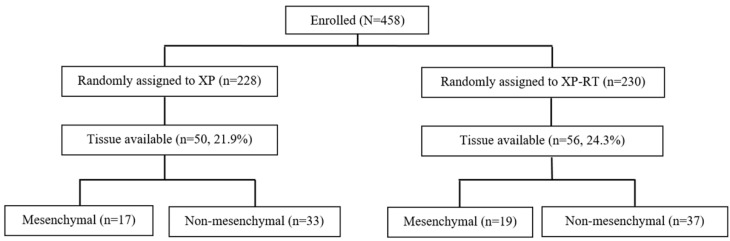
Flow diagram of patient inclusion.

**Figure 2 cancers-12-00943-f002:**
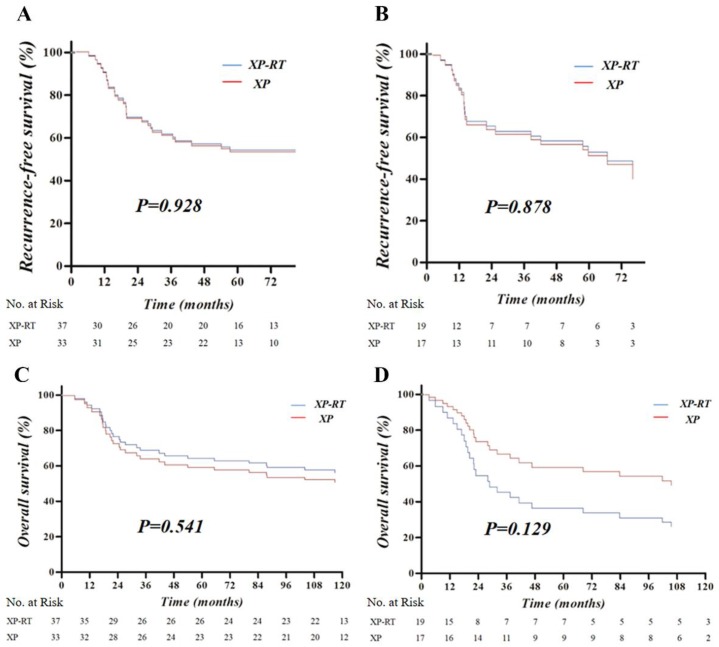
Adjusted curve of recurrence-free survival (RFS, **A**,**B**) and overall survival (OS, **C**,**D**) according to the use of adjuvant XP-RT or XP for the mesenchymal (**B**,**D**) and non-mesenchymal subtypes (**A**,**C**). In these curves which were displayed for the average value of the covariates in the study population, no significant differences related to the use of adjuvant XP-RT or XP were detected, although OS curves were inverted depending on adjuvant modality for the mesenchymal and non-mesenchymal subtypes (The presented p-values were based on HR test in the Cox proportional hazards model.).

**Figure 3 cancers-12-00943-f003:**
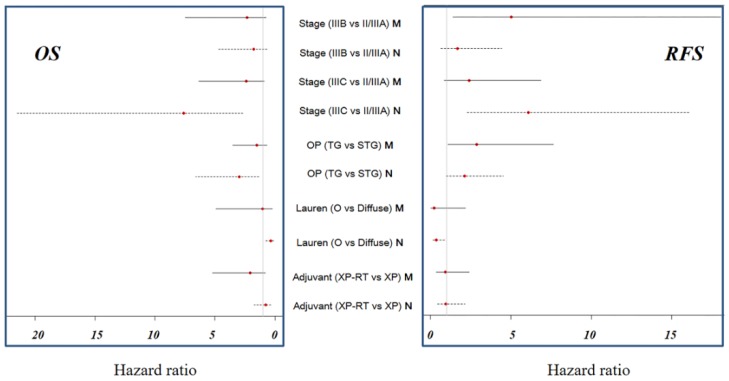
Forest plot for adjusted RFS and OS. No significant differences in RFS related to the use of adjuvant XP-RT or XP for the mesenchymal and non-mesenchymal subtypes were detected (OP, operation; TG, total gastrectomy; STG, subtotal gastrectomy; O, others, M, mesenchymal subtype; N, non-mesenchymal subtype)**.**

**Table 1 cancers-12-00943-t001:** Baseline characteristics of patients.

Variables	Mesenchymal (n = 36, %)	Non-Mesenchymal (n = 70, %)	*p*
Age—yr			0.185
Median	50	57	
Range	36–76	35–75	
Sex			0.019
Male	17 (47.2)	50 (71.4)	
Female	19 (52.8)	20 (28.6)	
Macroscopic type			0.039
0 (superficial)	1 (2.8)	3 (4.3)	
1 (mass)	0 (0.0)	1 (1.4)	
2 (ulcerative)	3 (8.3)	18 (25.7)	
3 (ulceroinfiltrative)	18 (50.0)	39 (55.7)	
4 (diffuse infiltrative)	14 (38.9)	9 (12.9)	
Location of primary tumor			0.161
Proximal	1 (2.8)	3 (4.3)	
Body	23 (63.9)	32 (45.7)	
Antrum	12 (33.3)	29 (41.4)	
Multiple/diffuse	0 (0.0)	6 (8.6)	
Type of operation			0.128
Total gastrectomy	20 (55.6)	28 (40.0)	
Subtotal gastrectomy	16 (44.4)	42 (60.0)	
Lauren classification			0.004
Diffuse	32 (88.9)	42 (60.0)	
Intestinal	2 (5.6)	25 (35.7)	
Unclassified	2 (5.6)	3 (4.3)	
No of dissected LNs			0.824
Median	42	43	
Range	18–92	14–96	
No of positive LNs			0.156
Median	10	8	
Range	0–50	1–38	
AJCC 8th stage (pathologic stage)			0.248
II	0 (0.0)	2 (2.9)	
IIIA	9 (25.0)	23 (32.9)	
IIIB	16 (44.4)	34 (48.6)	
IIIC	11 (30.6)	11 (15.7)	
LVI			0.402
Positive	29 (80.6)	55 (78.6)	
Negative	6 (16.7)	15 (21.4)	
Unknown	1 (2.8)	0 (0.0)	
PNI			0.724
Positive	21 (58.3)	42 (60.0)	
Negative	12 (33.3)	28 (40.0)	
Unknown	3 (8.3)	0 (0.0)	
ARTIST			0.994
XP-RT	19 (52.8)	37 (52.9)	
XP	17 (47.2)	33 (47.1)	

Abbreviations: LN: lymph node, AJCC: American Joint Committee on Cancer, LVI: lymphovascular invasion, PNI: perineural invasion, XP: capecitabine and cisplatin, XP-RT: XP-radiation therapy.

**Table 2 cancers-12-00943-t002:** Univariate analysis of loco-regional recurrence-free survival (LRRFS), recurrence-free survival (RFS), and overall survival (OS).

Variable	Factor	Reference	LRRFS			RFS			OS		
			*p*	HR	95% CI	*p*	HR	95% CI	*p*	HR	95% CI
Age			0.887	1.00	0.95–1.04	0.674	0.99	0.97–1.02	0.950	1.00	0.97–1.03
Sex	female	male	0.700	0.82	0.31–2.20	0.519	1.21	0.68–2.14	0.418	1.26	0.72–2.21
Macroscopic type	3	0/1/2	0.762	1.20	0.38–2.22	0.057	2.52	0.97–6.53	0.082	2.34	0.90–6.09
	4		0.137	2.94	0.71–12.10	<0.001	7.35	2.67–20.20	<0.001	6.78	2.50–18.40
Lauren type	others	diffuse	0.137	0.43	0.14–1.31	0.028	0.46	0.23–0.92	0.052	0.52	0.26–1.01
LVI	positive	negative	0.153	3.02	0.66–13.80	0.822	1.09	0.53–2.24	0.851	0.94	0.47–1.87
PNI	positive	negative	0.044	3.57	1.03–12.20	0.014	2.28	1.18–4.40	0.034	2.00	1.05–3.78
Type of operation	TG	STG	0.504	1.36	0.55–3.36	0.007	2.18	1.24–3.82	0.002	2.38	1.36–4.17
Stage	IIIB	II/IIIA	0.844	1.12	0.35–3.59	0.163	0.92	0.81–3.49	0.193	1.62	0.78–3.37
	IIIC		0.670	1.29	0.40–4.22	<0.001	2.91	1.55–5.49	<0.001	3.11	1.67–5.80
Subtype	mesenchymal	non-mesenchymal	0.275	1.71	0.65–4.46	0.009	2.11	1.21–3.70	0.003	2.28	1.31–3.96
ARTIST	XP-RT	XP	0.196	0.52	0.20–1.40	0.500	1.21	0.69–2.11	0.393	1.27	0.73–2.21

Abbreviations: LRRFS: loco-regional recurrence-free survival, RFS: recurrence-free survival, OS: overall survival, HR: hazard ratio, CI: confidence interval, LVI: lymphovascular invasion, PNI: perineural invasion, TG: total gastrectomy, STG: subtotal gastrectomy, XP: capecitabine and cisplatin, XP-RT: XP-radiation therapy.

**Table 3 cancers-12-00943-t003:** Multivariate analysis of recurrence-free survival (RFS), and overall survival (OS) in mesenchymal and non-mesenchymal subtypes.

			RFS			OS		
Variable	Factor	Reference	*p*	HR	95% CI	*p*	HR	95% CI
Mesenchymal subtype								
Stage	IIIB	II/IIIA	0.013	5.03	1.40–18.05	0.146	2.33	0.73–7.48
	IIIC		0.098	2.41	0.85–6.86	0.079	2.39	0.90–6.34
Type of operation	TG	STG	0.035	2.88	1.08–7.66	0.319	1.53	0.66–3.51
Lauren type	other	diffuse	0.202	0.24	0.03–2.17	0.981	1.02	0.21–5.93
ARTIST	XP-RT	XP	0.878	0.93	0.36–2.39	0.129	2.05	0.81–5.20
Non-mesenchymal subtype								
Stage	IIIB	II/IIIA	0.305	1.67	0.63–4.43	0.257	1.77	0.66–4.74
	IIIC		<0.001	6.08	2.28–16.19	<0.001	7.61	2.68–21.61
Type of operation	TG	STG	0.055	2.11	0.98–4.54	0.007	2.99	1.35–6.59
Lauren type	other	diffuse	0.031	0.36	0.15–0.91	0.029	0.34	0.13–0.89
ARTIST	XP-RT	XP	0.928	0.96	0.44–2.14	0.541	0.78	0.34–1.76

Abbreviations: RFS: recurrence-free survival, OS: overall survival, HR: hazard ratio, CI: confidence interval, TG: total gastrectomy, STG: subtotal gastrectomy, XP: capecitabine and cisplatin, XP-RT: XP-radiation therapy.

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
