# Peer review of "Outcomes of Radiotherapy for Mesenchymal and Non-Mesenchymal Subtypes of Gastric Cancer"

_cancers, 2020, doi:10.3390/cancers12040943_

Round 1

Reviewer 1 Report

In my opinion, the overall level of the paper is very good structured: it is well written and important considerations are highlighted. The discussion sections provide useful information for the readers and the conclusions appear rationale, emphasizing the need to perform further investigations in larger and homogeneous series to clarify the real effect of adjuvant XP-RT in mesenchymal and non-mesenchymal molecular subtypes of gastric carcinomas.

Author Response

Thank you for your insightful and generous comments.

Reviewer 2 Report

The authors of reviewed manuscript are assessing the outcomes of two approaches to the treatment of patients with gastric cancers – particularly mesenchymal subtypes. The materials and data for this study come form formerly conducted ARTRIST trial. Although the authors showed no major differences between achieved outcomes or did not show any major advantages of adjuvant XP-TR over XP, furthermore they showed some limitations of their study (with which I agree) – in my opinion this manuscript deserves to be published. This article is showing some direction of further broader studies and may be an advise in choice of therapies to the particular subtypes of gastric cancer.

Minor comments:

Some abbreviations are not explained in the abstract which could be discouraging for potential readers not necessarily related strongly with oncology. The same goes for over use of abbreviations in the text. Moreover some abbreviations are not explained in the text at the firs appearance, although explained in the abstract. Some abbreviations are explained in the tables captions, however there is lack of explanation in the text (at the firs appearance). Given the number of abbreviations used, the reader would understand the text more easily.

Lines 78-79 – the sentence not completed is a repetition of the next one.

In my opinion the introduction section lacks of the hypothesis under the study – please add one or two short sentences. The rest of this paragraph seems to suggest authors expectations.

Figures incorporated to the manuscript are hard to read due their resolution. I hope these are not the final figures.

Author Response

The authors of reviewed manuscript are assessing the outcomes of two approaches to the treatment of patients with gastric cancers – particularly mesenchymal subtypes. The materials and data for this study come from formerly conducted ARTRIST trial. Although the authors showed no major differences between achieved outcomes or did not show any major advantages of adjuvant XP-TR over XP, furthermore they showed some limitations of their study (with which I agree) – in my opinion this manuscript deserves to be published. This article is showing some direction of further broader studies and may be an advise in choice of therapies to the particular subtypes of gastric cancer.

→Thank you for your insightful and generous comments.

Minor comments:

Some abbreviations are not explained in the abstract which could be discouraging for potential readers not necessarily related strongly with oncology. The same goes for over use of abbreviations in the text. Moreover some abbreviations are not explained in the text at the first appearance, although explained in the abstract. Some abbreviations are explained in the tables captions, however there is lack of explanation in the text (at the first appearance). Given the number of abbreviations used, the reader would understand the text more easily.

→ Thank you for your thoughtful comments. We have added explanation for some abbreviations as your advice.

Lines 78-79 – the sentence not completed is a repetition of the next one.

→ Thank you for your comments. We changed the sentence.

In my opinion the introduction section lacks of the hypothesis under the study – please add one or two short sentences. The rest of this paragraph seems to suggest authors expectations.

→ We changed the paragraph as follows:

“Recently, our group performed a study to predict outcomes in patients with the mesenchymal subtype using a NanoString assay in 70 ACRG specimens [10]. The mesenchymal subtype showed significantly worse survival compared to the non-mesenchymal subtype following curative surgery in GC [11-13]. The impact of mesenchymal subtype which could be clearly related with radioresistance, like hypoxia and PARP-1, in terms of radiotherapy (RT) efficacy has not been defined yet although several studies have demonstrated that mesenchymal subtype predicts poor outcome following chemotherapy in GC [11-13]. It would be appropriate to evaluate the efficacy of RT on mesenchymal subtype of GC through the ARTIST trial which had been conducted to compare postoperative chemotherapy (XP) versus chemoradiotherapy (XP-RT) following complete curative resection with D2 lymph node dissection in GC (clinical trials.gov identifier NCT00323830).”

Figures incorporated to the manuscript are hard to read due their resolution. I hope these are not the final figures.

→ Thank you for your comments. We will add higher resolution figures separately.